# The Effects of Short-Term Immunosuppressive Therapy on Redox Parameters in the Livers of Pregnant Wistar Rats

**DOI:** 10.3390/ijerph16081370

**Published:** 2019-04-16

**Authors:** Dagmara Szypulska-Koziarska, Aleksandra Wilk, Joanna Kabat-Koperska, Agnieszka Kolasa-Wołosiuk, Jolanta Wolska, Barbara Wiszniewska

**Affiliations:** 1Department of Histology and Embryology, Pomeranian Medical University in Szczecin, Powst. Wlkp. 72, 70-111 Szczecin, Poland; dagmara.koziarska@pum.edu.pl (D.S.-K.); agnieszka.kolasa@pum.edu.pl (A.K.-W.); barbara.wiszniewska@pum.edu.pl (B.W.); 2Department of Nephrology, Transplantology and Internal Medicine, Pomeranian Medical University in Szczecin, Powst. Wlkp. 72, 70-111 Szczecin, Poland; askodom@poczta.onet.pl; 3Department of Biochemistry and Human Nutrition, Pomeranian Medical University in Szczecin, Broniewskiego 24, 71-460 Szczecin, Poland; jolanta.wolska@gmx.com

**Keywords:** oxidative stress, immunosuppressive drugs, liver, transplantation, pregnancy, redox balance

## Abstract

Immunosuppressive drugs are widely used to avoid graft rejection, but they are also known to be strongly hepatotoxic. The goal of the current study was to determine: (i) the immunoexpression of SOD1, CAT, GPX1; (ii) the concentration of MDA, GSH; (iii) the activity of SOD, CAT, GPX, in the native liver of a pregnant female rats undergoing immunosuppressive therapy. The study was based on archival material obtained from Department of Nephrology, Transplantology and Internal Medicine of the Independent Public Clinical Hospital No. 2 at the Pomeranian Medical University in Szczecin, Poland. The study was carried out on 32 female rats exposed to oral administration of immunosuppressants two weeks before and during pregnancy. The percentage of SOD1 immunopositive hepatocytes in rats treated with cyclosporine A, mycophenolate mofetil, everolimus, and glucocorticosteroid was significantly elevated above that of the control rats. The concentration of MDA in the liver of animals exposed to cyclosporine A, everolimus, and glucocorticosteroid was significantly higher than in other groups. Among the groups of dams treated with immunosuppressive drugs, the highest significant concentration of GSH was found in the livers of rats treated with cyclosporine A, mycophenolate mofetil and glucocorticosteroid. Immunosuppressive therapy during pregnancy affects the oxidoreductive balance in the livers of rats, depending on the regimen used.

## 1. Introduction

Following the transplantation of solid organs, the immune system of recipients engages in a range of mechanisms aimed at overcoming foreign agents, which can contribute to graft failure. The rejection of allografts is mediated primarily by infiltration of the graft by T lymphocytes and the range of inflammatory reactions that arise due to their presence. The proliferation of mononuclear leukocytes, such as lymphocytes and monocytes, is a clear demonstration of acute and chronic rejection within the allograft [1]. Immunosuppressive drugs are widely used to avoid graft rejection, but it is also well known that immunosuppressive drugs are strongly nephrotoxic and hepatotoxic [2,3]. 

Different immunosuppressants are prescribed in combinations of three drugs (each from a distinct group) after kidney transplantation. Calcineurin inhibitors (CNIs) are one of these groups, and include cyclosporine A (CsA) and tacrolimus (TAC) (Neuberger, 2004). Both of these very efficiently inhibit T-cell activation by binding to cyclophilin or immunophilin protein FKBP-12, forming specific complexes, CsA-cyclophilin and TAC-FKBP-12. These inhibit calcineurin phosphatase activity, thus preventing nuclear translocation on the nuclear factors of T-cells [4]. Moreover, TAC inhibits the transcription of cytokines gene, including interleukins and IFN in human T-cells [5,6]. Although CsA significantly improves graft survival, its use unfortunately induces several side effects. Hepatic dysfunction is a significant problem for some transplant recipients, and may lead to such conditions as hypertension, cholestasis, hypoproteinemia, hyperbilirubinemia, increased alkaline phosphatase and transaminase activity, and inhibition of protein synthesis [7]. The hepatic function may be affected even in the therapeutic range of CsA doses. Immunosuppressive therapy based on TAC can result in alopecia, post-transplant diabetes, and gastrointestinal system disorders. Other groups of immunosuppressive drugs are inhibitors of the inosine monophosphate dehydrogenase (IMDH inhibitors), including mycophenalate mofetil (MMF) [8,9]. Mycophenolate mofetil is added to the regimen, but it is worthy of mentioning that it is contraindicated during pregnancy, because it is a strong antiproliferating factor [10]. Similarly, the drugs known as mTOR inhibitors, which include everolimus (E), exhibit antiproliferative properties and are therefore contraindicated during pregnancy [11]. The last group of immunosuppressive drugs is the glucocorticosteroids (G). Glucocorticosteroids exhibit immunosuppressive and anti-inflammatory properties [12]. They inhibit the expression of genes for proinflammatory cytokines, such as interleukins 1 and 6. Another function of theirs is to decrease the activity of antigen-presenting cells, to induce apoptosis of lymphocytes, and to inhibit the diapedesis of leukocytes [13].

Although many studies have been carried out to elucidate the molecular mechanisms leading to the hepatotoxicity induced by immunosuppressive drugs, no satisfactory explanation has yet been arrived at. Most likely, several mechanisms are involved in immunosuppressant-induced hepatotoxicity, but some research has strongly suggested that the excessive production of reactive oxygen species (ROS) and the consequent imbalance between oxidants and endogenously produced antioxidants can lead to oxidative stress (OS) [7,14]. Currently, one of the most commonly assessed markers of OS is the concentration of malonyl dialdehyde (MDA), which is a product of lipid peroxidation [15]. Reactive oxygen species include a large group of single-electron, double-electron, and triple-electron reduction products of the oxygen molecule and singlet oxygen. This group contains the superoxide anion, ozone, hydrogen peroxide, the hydroxyl radical, alkyl radicals, and organic hydroxyl peroxides. The damage caused to molecules, cells, tissues, and organs depends on the nature of ROS activity, exposure time, temperature, partial pressure, and environmental factors [16]. Oxidative species are produced under physiological and pathological conditions in different cells, and are also released by the cells of the immune system [16]. In organism, the level of ROS is precisely controlled by an effective antioxidative system, including enzymes and nonenzymatic antioxidants. Superoxide dismutases (SODs), catalase (CAT), glutathione peroxidases (GPXs), glutathione reductase (GR), and glutathione-S-transferases (GSTs) constitute an enzymatic antioxidant defense system. In addition, reduced glutathione (GSH) is the most important cellular nonenzymatic antioxidant [12]. The activity level of these substances may vary according to severity of oxidative imbalance. GSH plays a crucial role in protecting cells from excessive ROS activity [16]. It has been reported that elevated concentrations of oxidized glutathione could lead to activation of various regulatory enzymes, and may be a cause of liver failure during immunosuppressive therapy [17]. 

As a detoxification organ, the liver is of utmost importance for the homeostasis for the entire body; it is also more prone to damage caused by numerous factors, including excessive levels of ROS [18]. The data obtained from numerous publications regarding the issue of the individual molecular mechanisms that lead to hepatotoxicity induced by each immunosuppressive drug, are still unclear. Far less is known about the influence of few different immunosuppressive medicines used as a co-therapy. Keeping in mind that different immunosuppressants can interfere with each other at the level of molecular mechanisms in both synergistic and antagonistic ways. The current paper is an attempt to broaden our knowledge of the effect of modern immunosuppressive regimens on the liver’s redox balance. Its aims were to determine:the immunoexpression levels of antioxidant enzymes (SOD1, CAT, GPX1) in the native liver of female rats under immunosuppressive therapy during pregnancy;the concentration of MDA, which is an indicator of OS;the concentration of GSH;the activity of SOD, CAT, and GPX.

## 2. Material and Methods

### 2.1. Animals and Treatment

This study is based on archival material obtained from the Department of Nephrology, Transplantology and Internal Medicine, Pomeranian Medical University in Szczecin, Poland. The current paper is a continuation of an experiment regarding the influence of immunosuppressive treatment on the morphology and apoptosis intensity in the native liver of pregnant Wistar rats (Szypulska-Koziarska et al. 2018). In brief: the experiment was conducted on 32 female Wistar rats and involved eight male Wistar rats for mating purposes only (all from the Centre for Experimental Medicine, Białystok Medical University, Poland). Initially, the rats were twelve weeks old and their mean weight was 230 g. All animals had genetic and health certificates issued by a veterinarian. This study was approved by the Szczecin Local Ethical Committee for Experiments on Animals (decision no. 12/2013, dated 24 Oct 2013). All the procedures involving animals were performed in accordance with the ethical standards of the institution at which the studies were conducted. The animals were housed individually on a 12-hour light–dark cycle and were fed with Labofeed H (Morawski, Kcynia, Poland) and water ad libitum. Prior to mating, the female rats were divided into 4 groups; one control, that did not receive any medicaments, and three experimental groups, that received the following immunosuppressive regimens: (i) CMG: cyclosporine A, mycophenolate mofetil, prednisone; (ii) TMG: tacrolimus, mycophenolate mofetil, prednisone; (iii) CEG: cyclosporine A, everolimus, prednisone. These drugs protocols were chosen to reflect the most common therapies in human transplant recipients. The experiment was performed using the pharmaceutical form of each drug. The animals received drugs through a stomach tube at a dose volume of 5 mL/kg b.w. every 24 h. The doses of medicaments were calculated to be equivalent to those used in human therapy. Furthermore, metabolic differences were also taken into consideration. The doses used in the study were as follows: 4 mg/kg b.w./day tacrolimus (Prograf, Astellas, Poland); 20 mg/kg b.w./day mycophenolate mofetil (CellCept, Roche, Poland); 5 mg/kg b.w./day cyclosporin A (Sandimmun Neoral, Novartis, Poland); 0.5 mg/kg b.w./day everolimus (Certican, Novartis, Poland); and 4 mg/kg b.w./day prednisone (Encorton, Polfa, Poland). The drug doses were based on data available in the literature [19,20,21,22,23,24] to reach a concentration within the therapeutic range. The study model is presented in Table 1. The animals received medication every 24 hours for approximately five weeks (two weeks prior to mating, when they were placed with males 1:1 in separate cages, and later in the third week of pregnancy). After mating, each pregnant female rat was housed in a separate cage. Once a week, the pregnant animals were weighed again, and the medication dose was adjusted to the actual weight. After delivery, the treatment was stopped. No drugs were administrated during lactation (breastfeeding is not advised while taking immunosuppressive drugs). Thirty-one female rats completed the study; 69 pups from the control group, thirteen from the CMG group, and only one pup from the CEG group were born. The dams were sacrificed at weaning, on day 21 after delivery by sodium pentobarbital (Polpharma, Poland) administered by intraperitoneal injection at a dose of 40 mg/kg body weight. Subsequently, necropsies of all rats were performed. The livers were collected, weighed, and partly fixed in 4% buffered formalin solution for immunohistochemical examination, and partly placed in liquid nitrogen and stored at −80 °C until analysis for markers of oxidative stress.

### 2.2. Immunohistochemistry

Immunostaining was performed on formalin-fixed, paraffin-embedded rat liver tissue sections. Sections 3 μm thick were initially deparaffinized with xylene and rehydrated in graded alcohols, followed by rinsing with distilled water. Antigen retrieval was carried out by microwaving the slides in 10 mM sodium citrate buffer (pH 6.0). Endogenous peroxidase activity was blocked with 3% H_2_O_2_. After slow cooling to room temperature, the slides were washed in PBS twice for 5 min. To determine the immunoexpression and localization of the antioxidant enzymes SOD1, CAT, and GPX1, the following antibodies were used: anti-SOD1 (rabbit polyclonal antibody, FL-154, Santa Cruz-1407);anti-CAT (rabbit polyclonal antibody, H-300, Santa Cruz-50508);anti-GPX1 (mouse monoclonal antibody, D-2, Santa Cruz-376877).

Incubation with the appropriate antibody lasted 60 min at room temperature, and the final dilution was 1:250. The primary antibodies were omitted in the negative control. Next, all slides were stained with a labeled polymer peroxidase system with 5,5’-diaminobenzidine (DAB) as the chromogen (EnVision + System-HRP, code K4010, DakoCytomation, Glostrup, Denmark) in accordance with manufacturer’s staining protocol. The sections were washed in distilled water and counterstained with hematoxylin. Positive staining was defined microscopically (Leica DM5000B, Wetzlar, Germany) by visual identification of brown pigmentation in the cytoplasm of the hepatocytes. The percentage of positively stained cells was calculated by the following steps: initially, all the hepatocytes from an equal area of the parenchyma of the liver (objective magnification ×100) were counted (A-parameter). Secondly, the number of positively stained hepatocytes was counted from the same area (B-parameter). In the next stage, the A-parameter was taken to represent 100%, and the B-parameter was calculated as the percentage of apoptotic hepatocytes. These steps were repeated for each group in six randomly chosen microphotographs from six different rats. 

### 2.3. Preparation of Liver Homogenates

The frozen livers were removed from the liquid nitrogen and placed in a thermobox at −21°C. A small fragment of the hepatic tissue was placed in a metal homogenizator (previously cooled in a container with liquid nitrogen) and liquid nitrogen was poured over it 2–3 times; it was then fragmented with 4–5 hammer blows against a metal mandrel (also previously cooled in a container using liquid nitrogen). The pulverized and frozen sample (approximately 1 mg of tissue) was placed using a chilled spoon in an Eppendorf tube containing 500 µL of appropriate buffer (according to the commercial enzyme assay kit’s procedure) that had been previously chilled to 4°C. After a short vortexation, homogenization was carried out with a knife homogenizator for about 15 s. Extract mixtures were centrifuged (3000 G for 10 min at 4 °C), and the supernatants were stored at −80°C and used for enzyme assays.

### 2.4. Determination of the Concentration of MDA: A Product of Lipid Peroxidation

The concentration of MDA, a lipid peroxidation (LPO) product, was measured using a Bioxytech LPO-586 Assay Kit (OxisResearch, Belgium), following the manual. Before the homogenization, 0.5 M butylated hydroxytoluene in acetonitrile was added to prevent oxidation of the sample during homogenization. PBS was used for sample homogenization. The LPO assay is based on reaction of chromogenic reagent N-methyl-2-phenylindole with MDA. In brief, 200 µL of sample was added to 650 µL N-methyl-2-phenylindole in acetonitrile. Next, 150 µL of 12 M HCl was added and samples were incubated at 45 °C for 60 min. After the incubation time, the samples were centrifuged, and the clear supernatant was transferred to cuvettes. The malonyl dialdehyde concentration was determined by measuring the absorbance at 586 nm, using an Alfa 40 spectrophotometer (PerkinElmer), and calibrated using standard curves.

### 2.5. Determination of the Activity of Enzymatic Antioxidants

#### 2.5.1. SOD Activity

Total (Cu-Zn and Mn) SOD (EC 1.15.1.1) activity was measured using a Bioxytech SOD-525 Assay Kit (OxisResearch, Poland), following the manual. The tissue was washed with 0.9% NaCl containing 0.16 mg/mL heparin before homogenization. PBS was used for sample homogenization. The assay is based on the SOD-mediated increase in the rate of autoxidation of 5,6,6a,11b-tetrahydro-3,9,10-trihydroxybenzo[c]fluorene in aqueous alkaline solution to yield a chromophore with a maximum absorbance at 525 nm. In brief, 40 µL of sample was incubated for one minute at 37 °C with 900 µL 2-amino-2-methyl-1,3-propanediol buffer (pH 8.8) containing boric acid and diethylenetriaminepentaacetic acid (DTPA), and with 30 µL 1-methyl-2-vinylpyridinium trifluoromethanesulfonate in HCl. Then 30 µL of 5,6,6a,11b-tetrahydro-3,9,10-trihydroxybenzo[c]fluorene was added and the reaction mixture was transferred to a cuvette. SOD activity was determined by measuring the absorbance at 525 nm, using an Alfa 40 spectrophotometer (PerkinElmer), over time. A 50% inhibition is defined as 1 unit of SOD and the activity was expressed as U/mg protein.

#### 2.5.2. CAT Activity

The activity of CAT (EC 1.11.1.6) was measured using a Bioxytech Catalase-520 Assay Kit (OxisResearch, Poland), following the manual. Phosphate saline buffer (PBS, 20 mM, pH 7.4) was used to homogenize the sample. Briefly, 30 µL of sample was incubated for one minute with 500 µL of 10 mM H_2_O_2_. The reaction was quenched using 500 µL sodium azide. Next, the reaction mixture (20 µL) was transfer to a cuvette and 2 mL of horseradish/chromogen reagent was added. Catalase activity was determined by measuring the absorbance at 520 nm, using an Alfa 40 spectrophotometer (PerkinElmer). Catalase activity was expressed as U/mg protein.

#### 2.5.3. GPX Activity

The activity of cellular GPX (EC 1.11.1.9) was measured using a Bioxytech GPX-340 Assay Kit (OxisResearch, Poland), following the manual. A buffer containing 1 mM mercaptoethanol, 5 mM EDTA, 50 mM TRIS-HCl was used to homogenize the samples. In brief, 350 µL of sample was added to 350 µL of 50 mM Tris-HCl buffer containing 0.5 mM EDTA, 5 mM NADPH, 42 mM reduced glutathione and 10 units/mL glutathione reductase. The reaction was initiated with the addition of 350 µL 30 mM tert-butyl hydroperoxide, and oxidation of NADPH was detected by monitoring the decrease in absorbance at 340 nm, using an Alfa 40 spectrophotometer (PerkinElmer). One unit of GPX activity was defined as the amount of sample required to oxidize 1 µM of NADPH per minute, based on the molecular absorbance of 6.22 × 106 for NADPH. GPX activity was expressed as U/mg protein.

### 2.6. Determination of GSH Concentration

The concentration of glutathione was measured using a Bioxytech GSH-400 Assay Kit (OxisResearch, Poland), following the manual. Ice-cold metaphosphoric acid was used to homogenize the samples. The samples were kept at −4 °C and used for GSH determination within one hour. The GSH assay is based on the reaction of 4-chloro-1-methyl-7-trifluromethyl-quinolinium methylsulfate with all mercaptans present in the sample. Next, the β-elimination reaction under alkaline conditions leads to the formation of a chromophoric thione with a maximal absorbance wavelength at 400 nm. In brief, 40 µL of sample was added to 860 µL of potassium phosphate containing diethylenetriamine pentaacetic acid (DTPA). Next, 50 µL of chromogenic reagent and 50 µL 30% NaOH were added, and the samples were thoroughly mixed and incubated at 25 °C for 10 min in the dark. GSH concentration was determined by measuring the absorbance at 400 nm, using an Alfa 40 spectrophotometer (PerkinElmer), and calculated using standard curves. 

### 2.7. Determination of Antioxidant System Efficiency

To evaluate the efficiency of antioxidant system within the cytoplasm and plasma membrane of hepatocytes, the respective SOD:CAT and SOD:GPX ratios were calculated. 

### 2.8. Statistical Analysis

The values of the quantitative variables were compared between groups using both parametric (Student’s *t*-test) and nonparametric tests (Mann–Whitney *U*-test). The parametric test was applied when the data were normally distributed, as assessed by the Shapiro–Wilk test. The nonparametric tests were used when the data were not normally distributed. The arithmetic mean (X), standard deviation (± SD), median, and minimum and maximum values were calculated for each group. The cut-off level of statistical significance was set at *p* < 0.05. Calculations were performed using Statistica 13.1 software (StatSoft, Kraków, Poland).

## 3. Results

The results of the study and the statistical analysis are presented in Figure 1 and Table 2, Table 3 and Table 4. 

### 3.1. Assessment of IHC Reaction to Localized Immunoexpression of SOD1, CAT, and GPX1

In the CMG and CEG groups, it was found that the percentage of SOD1 immunopositive hepatocytes was significantly higher than in the control group (Figure 1, Table 2). 

Only in the CEG group did the percentage of CAT immunopositive hepatocytes show statistically significantly higher values than both control and CMG groups (Figure 1, Table 2). 

In TMG and CEG groups, we observed that the percentage of GPX1 immunopositive hepatocytes was significantly higher than in the control group. Additionally, in the liver of CMG rats, the percentage of GPX1 immunopositive hepatocytes was significantly lower than in the TMG and CEG groups, while in TMG dams, the percentage of immunopositive hepatocytes was lower than in the CEG group (Figure 1, Table 2).

### 3.2. Assessment of the Enzymatic and Non-enzymatic Antioxidant System

#### Assessment of MDA and GSH Concentrations and of SOD, CAT, and GPX Activities

The concentration of MDA in the hepatic tissue of the CEG dams was significantly higher than in the control and CMG rats (Table 3).

There were no statistically significant differences in the liver activities of SOD or CAT among the experimental rats as compared to the control animals, and in each treatment group (Table 3).

We observed significantly lower activity of GPX in the hepatic tissue of TMG rats with a median value of 6.19, than in both the control and the CEG animals (Table 3). 

Among the groups of dams treated with immunosuppressive drugs, the highest statistically significant concentration of GSH was found in the CMG group, in comparison to the TMG and CEG groups. Additionally, in the TMG and CEG groups, the concentration of GSH was significantly lower than in the control group (Table 3). 

We noted significantly higher values of the SOD:GPX activity ratio in the hepatic tissue of TMG animals than in the control group. No significant differences were seen in the SOD:CAT activity ratio in any of the experimental groups (Table 4).

## 4. Discussion

The current paper is a continuation of an experiment regarding the influence of immunosuppressive treatment on the morphology and apoptosis intensity in the native livers of pregnant Wistar rats (Szypulska-Koziarska et al. 2018). In the aforementioned study, we observed numerous pathological alterations of hepatic tissue and increased apoptotic index of hepatocytes. Therefore, we decided to perform further examination to compare the impact of the most commonly used immunosuppressive regimens on the parameters of oxidative stress and antioxidant enzymes’ immunoexpression of hepatocytes. 

Immunosuppressive drugs are crucial for recipients who have received transplantations of solid organs, as they prolong the vitality of the graft, simultaneously prolonging the life of these patients. However, immunosuppressants are known for their toxicity, as they affect not only immunocompetent cells, but also other cells, such as mesenchymal cells, the endothelial cells of blood vessels, and other epithelial cells [25,26]. As the liver is an effective detoxification organ of great metabolic throughput, it is also believed to be one of the most vulnerable organs to the toxicity of immunosuppressive drugs, which can lead to, among other, oxidative stress (OS) [18]. This study has revealed that even short-term administration of cyclosporine A, everolimus, and glucocorticosteroid (CEG) leads to excessive lipid peroxidation in hepatocytes and, therefore oxidative stress. It was confirmed by significantly higher concentrations of MDA (a product of lipid peroxidation) in the liver of dams in CEG *vs* control and CMG groups. Malonyl dialdehyde is a common marker for OS, and we can therefore conclude that CEG regimen seems to be the most harmful from all drug protocols that were applied in current experiment. Additionally, MDA level in CMG group was significantly lower when compared to CEG, which was most probably due to the antioxidative properties of MMF [3,4]. The significantly elevated MDA level in the CEG group interferes with decreased level of non-enzymatic antioxidant, GSH, as was shown by us. The antioxidant protection in this group thus seems to be insufficient. Enhanced lipid peroxidation may result from decreased activity of GPX, and, indeed, we have observed here lower GPX activity vs the control, although this difference was not statistically significant. However, the immunohistochemical reaction showed a significantly increased percentage of GPX1 immunopositive hepatocytes. Notably, IHC confirms the localization of a particular proteins within the cell; however, it does not specify its activity. It is probably that, according to [27], although there were a high number of GPX1’ immunopositive hepatocytes, the GPX enzyme was mostly inactive. In addition, this can result from posttranslational modification of this protein. Because data on immunohistochemical analysis of the hepatocytes under immunosuppressive treatment is poor, it is problematic to compare our results with those of other studies. The biochemical assays of SOD and CAT activities in the livers of CEG animals did not show any significant differences compared to the control rats, unlike in other studies [28]. 

Most experiments on animals are based on monotherapy, so it is problematic to compare our results with those of other authors. However, we would like to emphasize that our study is based on combined therapy that reflects the transplant recipients’ therapy. Nonetheless, our results indicate that even short-term combined immunosuppressive therapy can affect the redox balance of the hepatic tissue in pregnant rats. In the present study, oxidative stress was manifested most intensively in CEG group. This group was the only one whose regimen included everolimus belonging to mTOR inhibitors. Kidney transplant recipients receiving mTORI, for instance everolimus, exhibited a greater risk of acute rejection among other non-mTORI users. Additionally, the use of mTORIs was associated with significantly increased risk of mortality versus non-mTORIs patients [29].

In the current study, immunohistochemical reaction in CEG rats revealed a greater number of SOD1 and CAT immunopositive hepatocytes than in the control rats. One explanation for this discrepancy might be that the amount of superoxide anions generated in the hepatocytes by the immunosuppressive drugs was not sufficiently high to raise SOD activity, even though the amount of the enzyme increased, as mentioned above. Our results are comparable with those of the study of Mostafavi-Pour et al. [28]. On the other hand, IHC method was firstly used to check the immunoexpression of SOD1 and CAT in hepatocytes, and secondly to localize the presence of these enzymes within the hepatocyte. As we noticed, increased IHC reaction was not reflected in the biochemical assay, most probably because these proteins were inactive due to post-translational modification [30]. 

In the liver tissue of dams treated with cyclosporine A, mycophenolate mofetil, and glucocorticosteroid (CMG) we did not confirm the appearance of oxidative stress, since the concentrations of MDA and GSH and the activities of SOD, CAT, and GPX enzymes were not significantly changed in comparison to the control group. Importantly, the mycophenolate mofetil is believed to exhibit an antioxidative properties [2,3], due to the fact that it can neutralize the free radicals. We can therefore conclude that the antioxidant defense of the hepatic tissue of dams treated with that particular regimen (CMG) was sufficient, and most likely, this drug protocol was sufficiently tolerated by the liver of pregnant rats that it did not induce excessive ROS formation. Considering IHC reaction, CMG animals, similar to the CEG group, showed a significantly higher percentage of SOD1 immunopositive hepatocytes than the control dams. Regarding the biochemical and IHC analyses, these CAT parameters correlate with each other, as we did not observe any differences in CAT activity or percentage of CAT’s immunopositive hepatocytes when compared with the control livers. Catalase is activated when the concentration of hydrogen peroxide is very high; that is why we did not observe excessive levels of MDA in the hepatic tissue of rats in CMG group. Therefore, our data may indicate that CMG regimen seems to be safe, especially in the consideration of pregnancy. 

In the current study, we did not confirm the presence of oxidative stress in the hepatic tissue of dams from the TMG group, since the concentration of MDA was not significantly elevated above the control rats. On the other hand, we noticed significantly decreased concentration of GSH and the activity of GPX in the livers of these rats, when compared with the control dams. We can suggest two explanations for these observations. Firstly, our experiment was based on short-term therapy, and although the concentration of MDA was not significantly elevated above the control, a rising trend of this process was clearly visible (0,97 nM/mg protein in the control vs 1,25 nM/mg protein in TMG). We suspect that long-term therapy may enhance the oxidative stress. Secondly, both TMG and CMG regimens include mycophenolate mofetil. Again, this particular immunosuppressive drug exhibits antioxidative properties [30], and that is perhaps why MDA level was not above the control level in these two groups yet. Additionally, the TMG group was the only one in which the activity of GPX was significantly lower than in the control rats, although IHC reaction indicated significantly higher percentages of GPX1 immunopositive hepatocytes than in either the control or the CEG group. It seems likely that, despite the immunosuppressants, there was no effect on translation of GPX1 gene, as GPX1 protein level was elevated; however, there could have been an adverse influence on post-translational modification, resulting in decreased GPX protein activity. Another explanation is that when GSH had been used up through neutralization of free radicals [31], the GPX activity, which strongly depends on the availability of GSH, decreased as well. Nonetheless, data on this effect are scarce, and this observation still requires an explanation. What is more, decreased level of GSH in TMG vs CMG can be explained by the fact that TAC exhibits more toxic properties than CsA [6].

Furthermore, the antioxidant defense of the cytoplasm of hepatocytes was efficient within all the groups, which was confirmed by rating the SOD:CAT ratio. On the other hand, the effectiveness of the antioxidant system in plasma membranes of hepatocytes was only efficient in CMG and CEG groups, but not in the TMG group, which was assessed by the SOD:GPX ratio.

In the early stages of an inflammatory disease, there is a homeostatic stimulation of the antioxidant defense system due to an increase in the amount of free radicals. Instead, soon after free radicals get to chronically elevated levels, this compensation ceases. For instance, in the early stages of coronary artery disease the amount of SOD and CAT become elevated to protect and prevent lipid peroxidation whereas they decrease significantly with the worsening of the disease [32].

Yet, in our study, we exposed pregnant dams to combined immunosuppressive therapy for 5 weeks only, which is classified as a short-term exposure, and therefore, we cannot consider its chronic effects.

## 5. Conclusions 

Based on our results, immunosuppressive therapy during pregnancy alters the redox balance in the livers of pregnant rats, in a manner dependent on the regimen employed. Our study indicates that:regimen based on CsA, E and G exhibits the most pro-oxidative properties from all of protocols used in current experiment,regimens including MMF seem to protect hepatic tissue against oxidative stress,TAC seems to be more hepatotoxic in case of redox parameters, than CsA.

## Figures and Tables

**Figure 1 ijerph-16-01370-f001:**
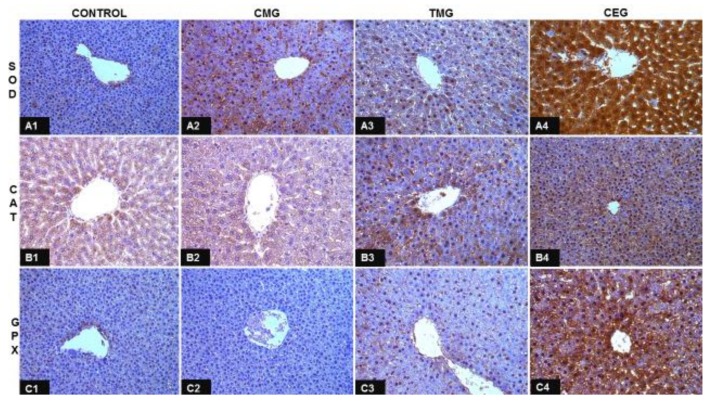
Representative images of livers from control rats (A1-C1) and rats from treatment groups (CMG: A2-C2; TMG: A3-C3; CEG: A4-C4) immunohistochemically stained to show immunoexpression of SOD1 (A2-A4), CAT (B2-B4) and GPX1 (C2-C4) in hepatocytes. Objective magnification 40×.

**Table 1 ijerph-16-01370-t001:** The study model.

Group	Glucocorticosteroids (G)	Tacrolimus (TAC)	Cyclosporin A (CsA)	Everolimus (E)	Mycophenolate mofetil (MMF)
Control group (*n* = 7)	—	—	—	—	—
CMG group (*n* = 8)	+	—	+	—	+
TMG group (*n* = 8)	+	+	—	—	+
CEG group (*n* = 8)	+	—	+	+	—

Abbreviations of drug names used as study group labels: CMG: CsA + MMF + G; TMG: TAC + MMF + G; CEG: CsA + E + G.

**Table 2 ijerph-16-01370-t002:** Percentage of SOD1, CAT, and GPX1 immunopositive hepatocytes in the liver of female rats in the control, CMG, TMG, and CEG groups.

Enzyme/Group		Control	CMG	TMG	CEG
**SOD1**	X ± SD	84.33 ± 13.65	94.61 ± 15.01	88.26 ± 24.2	98.66 ± 1.49
	Median	81	99.9	99.9	99
	Range	68–100	41–99.9	28–99.9	93–99.9
			*** *vs.* Control**		*** *vs.* Control**
**CAT**	X ± SD	26.67 ± 33.1	36.79 ± 10.5	64.84 ± 48.6	98.16 ± 5.57
	Median	11.5	38	99	99.9
	Range	0–73	20–51	0–99.9	76–99.9
					**** *vs.* Control**
					***** *vs.* CMG**
**GPX1**	X ± SD	8.05 ± 10.8	3.46 ± 2.14	57.23 ± 30.7	88.88 ± 11.41
	Median	1	3	78	91
	Range	1.0–33.0	1.0–9.0	11.0–90.0	60.0–100.0
			***** *vs.* TMG**	***** *vs.* Control**	***** *vs.* Control**
			**** *vs.* CEG**	***** *vs.* CEG**	

X: arithmetical mean; SD: standard deviation; p: level of significance; CMG: CsA + MMF + G; TMG: TAC + MMF + G; CEG: CsA + E + G. * *p* < 0.01 (Mann–Whitney *U*-test). ** *p* < 0.001 (Mann–Whitney *U*-test). *** *p* < 0.0001 (Mann–Whitney *U*-test).

**Table 3 ijerph-16-01370-t003:** MDA concentration, SOD, CAT, and GPX activity, and GSH concentration in the hepatic tissue of female rats of the control, CMG, TMG, and CEG groups.

Parameter/Group		Control	CMG	TMG	CEG
**MDA [nM/mg protein]**	X ± SD	0.97 ± 0.22	1.01 ± 0.15	1.25 ± 0.32	1.35 ± 0.23
	Median	1.01	1.07	1.29	1.42
	Range	0.59–1.27	0.76–1.13	0.85–1.77	1.06–1.69
					*** *vs.* Control**
					*** *vs.* CMG**
**SOD** **[U/mg protein]**	X ± SD	5.07 ± 1.86	5.99 ± 1.71	5.44 ± 1.54	4.92 ± 2.28
	Median	5.41	6.73	5.13	4.64
	Range	2.07–8.01	2.93–6.99	3.8–7.76	2.21–9.54
**CAT** **[U/mg protein]**	X ± SD	108.8 ± 6.17	110.82 ± 5.25	112.45 ± 2.3	110.11 ± 3.2
	Median	106.17	110.47	112.49	110.13
	Range	102.09–118.8	106.17–118.84	108.6–116.0	106.1–114.7
**GPX** **[U/mg protein]**	X ± SD	25.44 ± 20.61	11.03 ± 8.8	6.53 ± 2.97	18.24 ± 11.9
	Median	21.73	6.45	6.19	22.24
	Range	3.91–56.37	4.92–25.81	1.87–10.02	3.4–30.4
				*** *vs.* Control**	
				*** *vs.* CEG**	
**GSH** **[μM/mg protein]**	X ± SD	1.17 ± 0.26	1.16 ± 0.14	0.86 ± 0.11	0.93 ± 0.1
	Median	1.09	1.25	0.87	0.93
	Range	0.87–1.56	0.97–1.31	0.68–1.04	0.77–1.11
			**** *vs.* TMG**	*** *vs.* Control**	*** *vs.* Control**
			*** *vs.* CEG**		

X: arithmetical mean; SD: standard deviation; p: level of significance; CMG: CsA + MMF + G; TMG: TAC + MMF + G; CEG: CsA + E + G. * *p* < 0.05 (Student’s *t*-test). ** *p* < 0.001 (Student’s *t*-test).

**Table 4 ijerph-16-01370-t004:** The SOD:CAT and SOD:GPX activity ratios for hepatic tissue in the control, CMG, TMG, and CEG groups.

Parameters	Control	CMG	TMG	CEG
**SOD:CAT**				
**n**	7	5	7	8
**X ± SD**	0.047 ± 0.02	0.05 ± 0.01	0.05 ± 0.01	0.04 ± 0.02
**Median**	0.05	0.06	0.05	0.04
**Range**	0.02–0.08	0.03–0.06	0.03–0.07	0.02–0.09
**SOD:GPX**				
**n**	8	5	6	7
**X ± SD**	0.52 ± 0.67	0.75 ± 0.44	0.98 ± 0.56	0.76 ± 1.02
**Median**	0.26	0.556	0.79	0.17
**Range**	0.04–2.07	0.26–1.36	0.66–2.13	0.10–2.81
			*** *vs.* Control **	

* statistically significant with *p* < 0.05 (Mann–Whitney *U*-test). CMG: CsA + MMF + G; TMG: TAC + MMF + G; CEG: CsA + E + G.

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
