# Peer review of "The Effects of Short-Term Immunosuppressive Therapy on Redox Parameters in the Livers of Pregnant Wistar Rats"

_ijerph, 2019, doi:10.3390/ijerph16081370_

Round 1
Reviewer 1 Report
The manuscript entitled “The effects of short-term immunosuppressive therapy on redox parameters in the liver of pregnant Wistar rats” by Dagmara Szypulska-Koziarska, Aleksandra Wilk, Joanna Kabat-Koperska, Agnieszka Kolasa-Wołosiuk, Jolanta Wolska and Barbara Wiszniewska, describes studies where the hepatotoxicity of three different combinations of four immunosuppressive drugs (cyclosporine A (CsA), tacrolimus (T), mycophenalate mofetil (MMF) and glucocorticosteroids (G)) and everolimus (E)) commonly used to avoid graft rejection was investigated. For this the authors submitted 32 female rats to three different combinations (CMG: CsA + MMF + G; TMG: TAC + MMF + G; CEG: CsA + E + G) of the referred immunosuppressive drugs that were orally administrated two weeks before and during rat pregnancy. Published evidences suggest that toxicity/secondary effects of the individual immunosuppressive drugs may be a consequence of oxidative stress (OS) generation due to the production of reactive oxygen species (ROS), and the consequent imbalance between oxidants and endogenous antioxidants triggered by these drugs. This led the authors to investigate if the molecular mechanisms leading to the hepatotoxicity of the distinct combinations of the immunosuppressor drugs were due to ROS production. For this authors determined the MDA concentration, an indicator of oxidative stress in rats’ hepatocytes. By immunohistochemistry assays they have also evaluated the levels of the antioxidant enzymes SOD1, CAT, and GPX1 in hepatocytes. In parallel they used liver homogenates to determine the impact of the immunosuppressive drugs in the concentration of GSH and SOD, CAT, and GPX enzymes activities.
The study is pertinent, although essentially descriptive with poor explanation of the obtained results.
Specific comments:
· The major difficulties encountered by the authors to explain the obtained results seems to be the discrepancy between the immunocytochemistry and the biochemical data of enzyme activity; for example, the quantification of the number of stained hepatocytes for CEG and TMG combinations (except for CMG) shows higher numbers for SOD1, CAT and GPx1. However, the activities of SOD and CAT do not significatively change and GPx decreases. Interestingly MDA concentration slightly increases, and the variation is only significant for CEG combination.
The authors conclude that “Based on our results, immunosuppressive therapy during pregnancy alters the oxidoreductive balance in the livers of pregnant rats, in a manner dependent on the regimen employed” but they also state that “Concluding, it is very difficult to say precisely which regimen of immunosuppressive drugs is least harmful for patients. Further experimentations are needed to broaden our knowledge of the pathomorphology of the liver in females under immunosuppressive therapy, especially during pregnancy.” The two statements clearly show that the paper is fragile in their achievements and essentially the authors are not able to explain their apparently contradictory results.
· From what is available in literature the individual molecular mechanisms that underlie hepatotoxicity induced by each immunosuppressive drug, used in this study, is far from being completely understood, as also mentioned by the authors. This means that the performed study is ambitious since make comparisons between groups of immunosuppressive drugs from which nothing is said concerning their synergistic or even antagonistic actions at the level of molecular mechanisms. This poor characterization of the experimental system may lead to complex results with difficult interpretation.
· The authors evaluated the levels of the antioxidant enzymes SOD1, CAT, and GPX1 by counting the immunohistochemical positive hepatocytes. Although more biological material probably will be required, a more accurate quantification of the protein amount could be done by western blot.
· Concerning the observation that the number of hepatocytes positively stained for GPx clearly increases for example for CEG combination whereas GPx activity decreases. The authors refer that ” An explanation for this may be associated with our analysis methods, since we only assessed the presence of one isoform of GPX (the GPX1 isoform) with the IHC method, and our biochemical analysis was based on the activity of all isoforms of GPX. It seems possible that this might be a limitation of our study.”, this explanation does not make sense since if in the IHC method the antibody is only specific for the GFPX1 then they are under-evaluating the global amount of the all GPX isoforms, whereas they are evaluating the activity of all GPX isoforms since the method does not discriminate between them. However, they observe higher amounts and decreased activity. The observed apparent discrepancy is probably because not all the GPX molecules synthesized are active. The GPX activity may be inhibited by post-translational regulation. The authors refer this later in the discussion, but this observation requires additional investigation to consolidate the observations. This could be an important achievement for explaining the results. In literature are several evidences of how GPX1 may be post-translationally inhibited.
· Taking into account the “significantly higher concentrations of MDA does not increase in the liver of dams in the CEG group” the authors state that “The antioxidant protection thus seems to be insufficient; our study showed significantly lower levels of GSH in the hepatocytes of dams from the CEG group than in the control and CMG groups. Enhanced lipid peroxidation may result from decreased activity of GPX.”. If this is true why does MDA concentration not increase in TMG and CMG where GPX significantly decreases (more than in CEG)?
· Also, in literature there are observations that indicated that in the early stages of an inflammatory disease there is a homeostatic up-regulation of the antioxidant enzyme system in response to increased free radicals. As soon as free radicals get to chronically elevated levels, this compensation ceases. For example, in coronary artery disease in the early stages of CAD, SOD and CAT levels increased to protect and prevent lipid peroxidation whereas they decreased significantly with the worsening of the disease. The regulatory mechanisms and pathway vias underlying the antioxidant response may be complex when the different drug combinations are used.
It would be interesting to see the if the activities of the studied enzymes changed through the exposure period studied.
In the discussion authors can improve the analysis of their result if extend the bibliography to other scenarios were oxidative stress is triggered.
In conclusion in the present form the data are not consolidated and need to be more clarified before they transmit a clear message to the scientific community.
Author Response
Response to Reviewer 1 Comments
Dear Reviewer, we would like to thank you for helpful comments and criticism.
We believe that our revised manuscript is now more balanced and better represents our work. We hope that this revised manuscript is now acceptable for publication.
Point 1: The major difficulties encountered by the authors to explain the obtained results seems to be the discrepancy between the immunocytochemistry and the biochemical data of enzyme activity; for example, the quantification of the number of stained hepatocytes for CEG and TMG combinations (except for CMG) shows higher numbers for SOD1, CAT and GPx1. However, the activities of SOD and CAT do not significantly change and GPX decreases. Interestingly MDA concentration slightly increases, and the variation is only significant for CEG combination.
Response 1: Thank you for your meaningful comment, we are very grateful for that.
As far as the immunohistochemistry assay is concerned, in deed we have shown an increased number of SOD1’s, CAT’s and GPX1’s immunopositive positive hepatocytes in pregnant rats from TMG and CEG group, without any significant change in the activity of these enzymes.
Of note, IHC method confirms only the localization of a particular proteins, like SOD, CAT
or GPX, within the cell, however, it does not specify its activity. Therefore we can conclude, due to the biochemical study, that probably most molecules of these proteins (SOD, CAT, GPX) were inactive. And it can result from posttranslational modification of the proteins, what have been mentioned in the discussion. Thank you again we filled missing explanation of this mechanism within “discussion” section.
The authors conclude that “Based on our results, immunosuppressive therapy during pregnancy alters the oxidoreductive balance in the livers of pregnant rats, in a manner dependent on the regimen employed” but they also state that “Concluding, it is very difficult to say precisely which regimen of immunosuppressive drugs is least harmful for patients. Further experimentations are needed to broaden our knowledge of the pathomorphology of the liver in females under immunosuppressive therapy, especially during pregnancy.” The two statements clearly show that the paper is fragile in their achievements and essentially the authors are not able to explain their apparently contradictory results.
Response 1: Thank you for your comment. We have changed and added information
that explain, with more details, the mechanism that probably acted on hepatocytes.
Therefore, we also changed the conclusions.
Point 2: From what is available in literature the individual molecular mechanisms that underlie hepatotoxicity induced by each immunosuppressive drug, used in this study, is far from being completely understood, as also mentioned by the authors. This means that the performed study is ambitious since make comparisons between groups of immunosuppressive drugs from which nothing is said concerning their synergistic or even antagonistic actions at the level of molecular mechanisms. This poor characterization of the experimental system may lead to complex results with difficult interpretation.
Response 2: Thank you for your note, We are really appreciate it. The protocols
of immunosuppressive drugs we have used in our study were coordinated with medical doctors, which are specialist in the field of transplantology. The regimens of drugs were have chosen are nowadays commonly used in different clinical treatment depending on, among other,
the health status, potential pregnancy, active or past viral illness, the cancer. Following your suggestions we have broaden the information regarding the action of immunosuppressive drugs within “method” chapter.
Point 3: The authors evaluated the levels of the antioxidant enzymes SOD1, CAT, and GPX1 by counting the immunohistochemical positive hepatocytes. Although more biological material probably will be required, a more accurate quantification of the protein amount could be done by western blot.
Response 3: Thank you very much for this comment. We believe it would be of great importance to do the western blot assay. Unfortunately, our study was based on archival material and our access to the biological material was therefore limited. However,
after considering your suggestions will perform further analysis with usage of very small amount of biological material –Laser Capture Microdissection. However, we would like
to perform aforementioned analysis in the next, future part of our study, thus it requires much more time and, additionally, financial support.
Point 4: Concerning the observation that the number of hepatocytes positively stained for GPx clearly increases for example for CEG combination whereas GPx activity decreases. The authors refer that ” An explanation for this may be associated with our analysis methods, since we only assessed the presence of one isoform of GPX (the GPX1 isoform) with the IHC method, and our biochemical analysis was based on the activity of all isoforms of GPX. It seems possible that this might be a limitation of our study.”, this explanation does not make sense since if in the IHC method the antibody is only specific for the GFPX1 then they are under-evaluating the global amount of the all GPX isoforms, whereas they are evaluating the activity of all GPX isoforms since the method does not discriminate between them. However, they observe higher amounts and decreased activity. The observed apparent discrepancy is probably because not all the GPX molecules synthesized are active. The GPX activity may be inhibited by post-translational regulation. The authors refer this later in the discussion, but this observation requires additional investigation to consolidate the observations. This could be an important achievement for explaining the results. In literature are several evidences of how GPX1 may be post-translationally inhibited.
Response 4: We wish to thank you for your meaningful note. According to the discrepancy between the IHC and biochemical assays, as we have mentioned the IHC method is used
to check the immunoexpression and localization of GPX1 protein in hepatocytes. We confirmed significantly elevated GPX1’s immunoexpression in hepatocytes in pregnant rats under immunosuppressive therapy, which, unfortunately was not confirmed by biochemical assay. The most probably reason for that is that most molecules of GPX1 protein which have been translated within the cell were inactive, and, it was perhaps due to posttranslational modification of this particular protein. As you have mentioned in your comments, there are several evidences in the literature that explain how GPX1 may be post-translationally inhibited, therefore thank you to your comments. We have emphasized this in the” discussion” section and we have added the proper literature.
Point 5: Taking into account the “significantly higher concentrations of MDA does not increase in the liver of dams in the CEG group” the authors state that “The antioxidant protection thus seems to be insufficient; our study showed significantly lower levels of GSH in the hepatocytes of dams from the CEG group than in the control and CMG groups. Enhanced lipid peroxidation may result from decreased activity of GPX.”. If this is true why does MDA concentration not increase in TMG and CMG where GPX significantly decreases (more than in CEG)?
Response 5: Thank you very much for this constructive comment. It is very important.
Firstly, only in CEG group, rat were exposed to everolimus, whereas in CMG and TMG instead mycophenolate mofetil was included. Of note, mycophenolate mofetil is believed to possess antioxidative properties and that can explain why in these two groups (CMG and TMG)
the MDA concentration was not significantly increased. On the other hand, our study was based on short-term therapy, and it only lasted for 5 weeks. It seems possible, that if the experiment lasted for a longer time, the antioxidant defence in hepatic tissue of CMG and TMG groups would not be sufficient enough and MDA would become significantly elevated as it was
in CEG rats. So far only a trend of increasing concentration of MDA in CME and TMG groups is visible. Following your suggestion and advise we have added the missing explanation making the article more clear.
Point 6: Also, in literature there are observations that indicated that in the early stages of an inflammatory disease there is a homeostatic up-regulation of the antioxidant enzyme system in response to increased free radicals. As soon as free radicals get to chronically elevated levels, this compensation ceases. For example, in coronary artery disease in the early stages of CAD, SOD and CAT levels increased to protect and prevent lipid peroxidation whereas they decreased significantly with the worsening of the disease. The regulatory mechanisms and pathway vias underlying the antioxidant response may be complex when the different drug combinations are used.
It would be interesting to see the if the activities of the studied enzymes changed through the exposure period studied.
In the discussion authors can improve the analysis of their result if extend the bibliography to other scenarios were oxidative stress is triggered.
Response 6: We really appreciate your meaningful comments, thank you for that.
We have added the missing information, that you have suggested in the discussion.
Perhaps, as you have mentioned, the usage of the regimen based on different immunosuppressive drugs led to complexes regulatory mechanisms. Of note, the results of our experiment were different to these available in literature, firstly because our study was not
a monotherapy, and, secondly it was based on short-term therapy.
According to your suggestions, it would be interesting to see the if the activities
of the studied enzymes changed through the exposure period studied, however, unfortunately our experiment was based on archival material and due to that it seems to be impossible in this moment. What is more, the activities of antioxidant enzymes were evaluated in the hepatic tissue, and that is why we could obtain that biological material only once.
We have also improved the analysis of our result by extending the bibliography,
as you have suggested.
Reviewer 2 Report
Title: The effects of short-term immunosuppressive therapy on redox parameters in the liver of pregnant Wistar rats
Recommendation: Accept after minor revisions
The current manuscript titled “The effects of short-term immunosuppressive therapy on redox parameters in the liver of pregnant Wistar rats” determines the:
· immunoexpression of SOD1, CAT, GPX1;
· the concentration of MDA, GSH;
· the activity of SOD, CAT, GPX, in the native liver of pregnant female rats undergoing immunosuppressive therapy.
This is an important field of study given the overall burden of immunosuppressive therapies. The manuscript has been written very well and can be accepted in its current form with minor corrections.
Minor comments:
· In the Introduction section please include a reference after these sentences: “Immunosuppressive therapy based on TAC can result in alopecia, post-transplant diabetes, and gastrointestinal system disorders. Other groups of immunosuppressive drugs are inhibitors of the inosine monophosphate dehydrogenase (IMDH inhibitors), including mycophenalate mofetil (MMF)”.
· “Glucocorticosteroids exhibit immunosuppressive and anti-inflammatory properties”.
· “Currently, one of the most commonly assessed marker of OS is the concentration of malonyl dialdehyde (MDA), which is a products of lipid peroxidation”.
· “Superoxide dismutases (SODs), catalase (CAT), glutathione peroxidases (GPXs), glutathione reductase (GR), and glutathione-S-transferases (GSTs) constitute an enzymatic antioxidant defense system, and reduced glutathione (GSH) is the most important cellular nonenzymatic antioxidant”.
· In Figure 1 each image should be individually labeled (For example A, B, C…etc).
Author Response
Response to Reviewer 2 Comments
Dear Reviewer, we would like to thank you for helpful comments and criticism.
We believe that our revised manuscript is now more balanced and better represents our work. We hope that this revised manuscript is now acceptable for publication.
Point 1: In the Introduction section please include a reference after these sentences:
· “Immunosuppressive therapy based on TAC can result in alopecia, post-transplant diabetes, and gastrointestinal system disorders. Other groups of immunosuppressive drugs are inhibitors of the inosine monophosphate dehydrogenase (IMDH inhibitors), including mycophenalate mofetil (MMF)”.
· “Glucocorticosteroids exhibit immunosuppressive and anti-inflammatory properties”.
· “Currently, one of the most commonly assessed marker of OS is the concentration of malonyl dialdehyde (MDA), which is a products of lipid peroxidation”.
· “Superoxide dismutases (SODs), catalase (CAT), glutathione peroxidases (GPXs), glutathione reductase (GR), and glutathione-S-transferases (GSTs) constitute an enzymatic antioxidant defense system, and reduced glutathione (GSH) is the most important cellular nonenzymatic antioxidant”.
Response 1: Thank you for your meaningful note, we have added all the references you have suggested.
Point 2: In Figure 1 each image should be individually labeled (For example A, B, C…etc).
Response 2: Thank You, we appreciate your note and we followed your suggestion.
All the images are renamed, individually labeled, to facilitate the readers receiving
and understanding the issue.
Round 2
Reviewer 1 Report
Dear Sirs,
The fragilities previously identified in this work are still unsolved. However, the authors made efforts to improve discussion and to clarify the scientific message of their work.
Since the results are interesting and may trigger new research concerning immunosuppressive therapy agents and their relationship with oxidative stress, the manuscript deserves to be published.
English language needs to be revised/improved throughout the manuscript.
1-Please clarify this sentence:
The most severe oxidative stress was noticed in CEG group. Our results seem to interfere with other studies [29].
2-On the other hand, we have noticed significantly decreased concentration of GSH and the activity of GPX in the livers of these rats, when compared with the control dams. This issue can be twofold explained.
It is better saying something like:
We can suggest two explanations for these observations (…)
Yours sincerely,
Helena Soares
Author Response
Response to Reviewer 1 Comments (Round 2)
Dear Reviewer, thank you again for your comment. We have improved English language and have changed the sentences according to your suggestions. We hope that following version of the manuscript is, thanks to your comments, more valuable and understandable.
Point 1: Please clarify this sentence:
1-The most severe oxidative stress was noticed in CEG group. Our results seem to interfere with other studies [29].
2-On the other hand, we have noticed significantly decreased concentration of GSH and the activity of GPX in the livers of these rats, when compared with the control dams. This issue can be twofold explained.
Response 1: Thank You, we appreciate your note and we followed your suggestion. Therefore we have rephrased the abovementioned sentences into:
1- Currently, oxidative stress was manifested most intensively in CEG group. This group was the only one, whose regimen included everolimus belonging to mTOR inhibitors.
2- We can suggest two explanations for these observations.